# Thiophenium Salts as New Oxidant for Redox Polymerization under Mild- and Low-Toxicity Conditions

**DOI:** 10.3390/molecules28020627

**Published:** 2023-01-07

**Authors:** Alexis Barrat, Frédéric Simon, Jérôme Mazajczyk, Bruno Charriere, Stéphane Fouquay, Jacques Lalevee

**Affiliations:** 1Université de Haute-Alsace, CNRS, IS2M UMR 7361, F-68100 Mulhouse, France; 2Université de Strasbourg, F-67081, France; 3Bostik Smart Technology Centre, F-60280 Venette, France; 4Arkema, F92700 Colombes, France

**Keywords:** peroxide-free, redox polymerization, free radical polymerization, thiophenium salts

## Abstract

In mild conditions (under air, room temperature, no monomer purification and without any energy activation), redox free radical polymerization (RFRP) is considered as one of the most effective methods to polymerize (meth)acrylate monomers. In the past several years, there has been a growing interest in research on the development of new redox initiating systems (RISs), thanks mainly to the evolution of toxicity labeling and the stability issue of the current RIS based on peroxide and aromatic amine. In this study, a new, low-toxicity RIS based on thiophenium salt as the oxidant species is presented with various reductive species. The reactivity and the stability of the proposed RISs are investigated and the synthesis of new thiophenium salts reported.

## 1. Introduction

Since the discovery of redox initiating systems (RISs) by German researchers trying to improve thermal initiating systems using peroxide as a thermal accelerator [1], studies have been carried out to optimize this new method of polymer synthesis.

In this context, new RISs were discovered. Most of these new systems rely on low-energy bond dissociation, such as the O–O bond (peroxide) or the S–S bond (disulfide) with various reducing agents (metallic ions, thiols, carboxylic acids, amines, etc.), as the oxidizing agent [2]. Among these possible oxidizing agents, dibenzoyl peroxide (BPO) has been extensively studied for its ability to work with a wide panel of reducing agents in both aqueous and nonaqueous media [2].

Redox free radical polymerization (RFRP) using an RIS is one of the mildest methods to reduce energy consumption in polymer synthesis [3,4]. Indeed, no additional stimuli are needed to trigger the initiation step of the polymerization. In addition, RFRP can be performed in mild conditions [5,6,7,8] (under air, no further purification of the monomers and carried out at room temperature). The general mechanism of RFRP is illustrated in Figure 1.

The polymerization of monomers by using the RFRP method is based on the mixing of an oxidizing agent and a reducing agent, which are able to generate in situ radical species through a redox mechanism (Figure 1, K_r_, generation of radical species).

The radicals generated from the redox reaction (R^•^) may react with a monomer (M), leading to a propagating polymeric radical (Figure 1, K_i_, initiation step). This small polymeric radical is then able to react with other monomer units to yield a longer polymeric radical (Figure 1, K_p_, propagation step). Finally, the polymerization stops when a polymeric radical reacts with another macroradical in combination or through disproportionation (Figure 1, K_t_, termination step).

The most common RIS used at the academic and industrial scales is based on dibenzoyl peroxide and an amine compound. Dibenzoyl peroxide is used as the oxidizing agent for the reasons mentioned above, and the amine compound is widely used as a reductive reagent thanks mostly to its (i) low cost, (ii) high reactivity and (iii) possibility to polymerize in solvent-free conditions (bulk), which is more challenging for other reductive reagents, such as metallic complexes (solubility issue) [9]. Studies conducted on the amine structure revealed the criteria needed to optimize the efficiency of the RIS [9]: (i) the primary and secondary amines classes are not suitable for RFRP, because proton transfer is more favorable, resulting in low yields in initiating radicals; (ii) among the tertiary amines class, aromatic amines with the minimal steric hindrance have the best efficiency, but they also have higher toxicity.

Thus, the most widely used RIS at the industrial scale is based on a mixture of dibenzoyl peroxide (BPO) and a tertiary aromatic amine such as 4-*N,N*-Trimethylaniline (4-*N,N*-TMA) (Figure 1), which fulfills all the criteria mentioned above.

Even though the system is very effective, it has some drawbacks, such as the low stability of peroxide in monomers [10] and the relative toxicity of the aromatic amine class [11]. The toxicity issue of the aromatic amine class may be overcome by playing on the chemical structure of the amine. Indeed, 2-(*N*-methyl-p-toluidino)ethanol, for example, is nontoxic and yields similar results compared with 4-*N,N*-TMA. However, the instability of BPO in monomers is a dead end. Industrial formulations use BPO in an inert plasticizer to avoid side reactions leading to instability. Given the major disadvantage of inert plasticizers in formulations (unreacted material in the final polymer altering the mechanical properties), new studies must be carried out to propose alternative redox agents that are stable in monomers and nontoxic.

To replace the oxidizing agent (e.g., BPO), this work will focus on thiophenium salts as new oxidant species. These salts have been used in organic chemistry but only for trifluoromethylation reactions [12]. Our interest in these salts emerged from the structural similarity with sulfonium salts such as bis[4-(diphenylsulfonio)phenyl]sulfide and bis(hexafluoroantimonate). Onium salts such as sulfonium and iodonium salts are well known in the literature to generate radical species upon irradiation or heat [2,13]. Some RISs were actually designed using iodonium salt as an oxidizing agent. [14,15]. In order to fully characterize the reactivity of thiophenium salts, this study was conducted using (i) different (meth)acrylate monomers (Figure 2), (ii) different reducing agents (Figure 3) and (iii) different thiophenium salts (Figure 4). The development of new RIS is a huge challenge [16,17] that can also require the synthesis of new compounds [18].

## 2. Results and Discussion

### 2.1. Synthesis of Thiophenium Salts

#### 2.1.1. Synthesis of 3,7-Ditertiobutyl-5-(trifluoromethyl)dibenzothiophenium Trifluoromethanesulfonate (Thiophenium (I))

In a 25 mL two-neck round bottom flask, 0.5 g of sodium triflinate (1 eq., 3 mmol) are added under anhydrous and argon atmosphere conditions, followed by 10 mL of anhydrous nitromethane as solvent. After total dissolution, 1 mL of triflic anhydride (2 eq., 6 mmol) is added. After stirring for 10 min, the 4,4′-di-*tert*-butylbiphenyl (1 eq., 3 mmol, 0.799 g) is added as a solution in 5 mL of nitromethane. The reaction is monitored by TLC using cyclohexane as eluent.

After 24 h, the nitromethane is distilled off under reduced pressure and washed several times with toluene (4 × 3 mL) to remove as much nitromethane as possible.

The crude is then diluted in 3.5 mL of distilled water, followed by 3.5 mL of diethyl ether, causing the precipitation of the product after stirring overnight.

After filtration and several washings with diethyl ether (4 × 3 mL), the product is dried under reduced pressure at 40 °C, yielding a white solid (0.167 g, 21% yield).

#### 2.1.2. Synthesis of 2,8-Difluoro-S-(trifluoromethyl)dibenzothiophenium Hexafluorophosphate (Thiophenium IV)

In a 10 mL two-neck round bottom flask, 0.192 g of sodium hexafluorophosphate (1 eq., 1.14 mmol) is added, followed by 5 mL of acetonitrile as the solvent. After complete dissolution, 0.5 g of 2,8-difluoro-5 (trifluoromethyl)dibenzothiophenium trifluoromethanesulfonate (Thiophenium (II)) (1 eq., 1.14 mmol) is added. The mixture is heated at 60 °C for 2 h.

After evaporation of acetonitrile under reduced pressure, the crude is dissolved in 10 mL of dichloromethane and washed with a saturated solution of NaHCO_3_ (3 × 5 mL). The dichloromethane is distilled off; 5 mL of acetonitrile is added; and a large excess of diethyl ether is added until precipitation of the product occurred (approx. 20 mL). The product is obtained as a white powder (0.413 g, 83%).

### 2.2. Thiophenium Salts in Redox Systems

#### 2.2.1. Effect of the Reducing Agent

Thiophenium salts and reducing agents alone are stable in (meth)acrylic monomers. Polymerization occurs only by mixing the two formulations (cartridges), which clearly highlights a radical generation through the reaction between thiophenium salts and reducing agents.

Thiophenium (III) is used in combination with different reducing agents, presented in Figure 3, in the monomer (5-ethyl-1,3-dioxan-5-yl)methyl acrylate (EDMA). The efficiency of the polymerization is followed by optical pyrometry (Figure 5).

Thiophenium (III)/reduction agents show approximately similarly short gel times (Figure 5: 16 s, 11 s, 35 s and 32 s, curves 1, 2, 3 and 4, respectively), which could indicate a high yield in radical generation for all tested systems. The maximum temperature reached is different (Figure 5: 53 °C, 68 °C, 88 °C and 146 °C, curves 1, 2, 3 and 4, respectively), which could indicate higher final acrylate function conversions for the systems with higher temperatures reached.

The resulting polymers obtained from the system’s Thiophenium (III)/AEAE and Thiophenium (III)/TEMED (Figure 5: curves 1 and 2, respectively) present tacky surfaces (i.e., poor polymerization on the surface from high oxygen inhibition), whereas polymers obtained from Thiophenium (III)/DHPP and Thiophenium (III)/Na-Tol-sulfinate (Figure 5: curves 3 and 4, respectively) present tack-free surfaces (i.e., full polymerization on the surface and low oxygen inhibition).

Finally, compared with BPO/4-*N,N*-TMA, all Thiophenium (III) systems with different reducing agents yield better results, namely a faster gel time, a higher exothermic peak (Figure 5: curve 5 compared with curves 1, 2, 3 and 4) and better polymer appearance (gel-like polymer for reference BPO/4-*N,N*-TMA with a very poor surface curing).

This first study on the nature of the reducing agent shows excellent results for reducing agents selected from aromatic compounds (i.e., electron-rich compounds). For the remaining work, Na-Tol-sulfinate was used instead of DHPP, for availability reasons.

#### 2.2.2. Effect of the Oxidizing Agent

The reactivity of several thiophenium salts is investigated in the monomer EDMA using Na-Tol-sulfinate as the most effective reducing agent identified in Section 2.2.1 (Figure 6).

Thiophenium (III)/Na-Tol-sulfinate compared with Thiophenium (II)/Na-Tol-sulfinate (Figure 7: curves 3 and 2, respectively) shows higher reactivity (31 s vs. 52 s for gel time and 104 °C vs. 72 °C for maximum temperature reached). This may be due to the mesomeric effect of fluorine [19], which leads to the electroenrichment of the sulfur and thus to a less-potent oxidizer agent.

Thiophenium (III)/Na-Tol-sulfinate compared with Thiophenium (I)/Na-Tol-sulfinate (Figure 6: curves 3 and 1, respectively) shows a shorter gel time (31 s vs. 91 s) but a lower maximum temperature (104 °C vs. 111 °C). This may be explained by impurity traces from the Thiophenium (I) synthesis. Another explanation is the difference in molar contents; indeed, the molecular weight differences between Thiophenium (III) (402.3 g.mol^−1^) and Thiophenium (I) (514.3 g.mol^−1^) (so, 1%_wt_ of Thiophenium (I) are that the former contains less quantity (in mol) than with Thiophenium (III)) compared with the others thiophenium salts (Thiophenium (II): 438.3 g.mol^−1^ and Thiophenium (IV): 434 g.mol^−1^).

Thiophenium (II)/Na-Tol-sulfinate compared with Thiophenium (IV)/Na-Tol-sulfinate (Figure 6: curves 2 and 4, respectively) shows lower reactivity (52 s vs. 21 s for gel time and 72 °C vs. 98 °C for maximum temperature). This is due to the effect of the counter ion, as described by the hard and soft acids and bases (HSAB) theory [20]. Indeed, the PF_6_^−^ counter ion is considered as soft, whereas CF_3_SO_3_^−^ is listed as hard, resulting in a higher ion-dissociation rate for the PF_6_^−^ counter ion and thus a higher reactivity.

The study of the chemical structure of the thiophenium itself and its associated counter ion revealed (i) better reactivity with soft counter ions (according to the HSAB theory), (ii) better solubility of the thiophenium salts in a monomer with lipophilic substituents (*tert*-butyl groups) and (iii) lower reactivity with the electroenrichment of the sulfur.

#### 2.2.3. Effect of the Concentration

Thiophenium salts are not commercially available in large quantities. In order to find the optimum usage, the effect of the thiophenium salt concentration was studied, using Na-Tol-sulfinate as the reducing agent with EDMA as a monomer (Figure 7). The RIS at 1%_wt_ (29 s gel time, 146 °C maximum temperature reached) compared with the RIS at 0.25%_wt_ (58 s gel time, 122 °C maximum temperature reached) showed that the concentration can be significantly decreased (Figure 7: curves 1 and 2, respectively).

However, the RIS at 0.1%_wt_ (159 s gel time, 72 °C maximum temperature reached) and the RIS at 0.01%_wt_ (no polymerization occurs) showed that the concentration of the initiating system cannot be lowered too much (Figure 7: curves 3 and 5, respectively).

Nonetheless, working with dissymmetric concentrations in each cartridge with Thiophenium (III) 0.1%_wt_/Na-Tol-sulfinate 0.25%_wt_ (107 s gel time, 103 °C maximum temperature reached) yielded satisfying results compared with Thiophenium (III) 0.1%_wt_/Na-Tol-sulfinate 0.1%_wt_ (159 s gel time, 72 °C maximum temperature reached) (Figure 7: curves 4 and 3, respectively).

#### 2.2.4. Effect of the Monomer

The respective reactivities of the various monomers were compared (Figure 2) to determine the best RIS (e.g., Thiophenium (III)/Na-Tol-sulfinate) (Figure 8).

The RIS in EDMA (29 s gel time, 146 °C maximum temperature) compared with the RIS in GFMA (176 s gel time, 88 °C maximum temperature) (Figure 8: curves 1 and 2, respectively) showed a large difference in reactivity between the acrylate and methacrylate monomers [21].

The RIS in isobornyl acrylate (Figure 8: curve 3, no polymerization) highlighted the lack of solubility of thiophenium and sulfinate salts in a low-polar monomer. To improve this solubility, blends of EDMA and IBA were tested (Figure 8: curve 4, blend of 50%_wt_ EDMA and 50%_wt_ IBA, gel time of 131 s for 132 °C maximum temperature; Figure 8: curve 5, blend of 30%_wt_ EDMA and 70%_wt_ IBA, gel time of 312 s for 98 °C maximum temperature). However, the reactivity is lower thanks to the limited solubility of the RIS in these monomer blends.

As mentioned in Section 2.2.2., the replacement of the CF_3_SO_3_^−^ counter ion by PF_6_^−^ drastically increased the reactivity of the system, as shown in Figure 9 (gel time of 84 s and maximum temperature reached of 75 °C for Thiophenium (IV) vs. no polymerization for Thiophenium (II)—curves 2 and 1, respectively).

#### 2.2.5. Effect of Light Activation

For the different RISs based on Thiophenium (III)/Na-Tol-sulfinate (Figure 3), the conversion of (meth)acrylate functions were followed by RT-FTIR with and without light irradiation (Figure 10). The conversion of (meth)acrylate functions is greatly improved by light excitation (Figure 10B: curve 1 (0% of acrylate function conversion without light) vs. curve 2 (68% of acrylate function conversion with light excitation); Figure 10C: curve 1 (72% of methacrylate conversion without light) vs. curve 2 (84% of methacrylate conversion with light excitation)).

Also, the rate of conversion seems to be improved (i.e., the slope of the curves) (Figure 10C: curve 1 (72% of methacrylate function conversion without light after 283 s) vs. curve 2 (80% of methacrylate function conversion with light excitation for the same time), as an example). Nonetheless, the impact of light irradiation remains unclear because the interaction of light in EDMA monomer results in a lower rate of acrylate function conversion (Figure 10A: curve 1 (72% of acrylate function conversion without light after 62 s) vs. curve 2 (37% of acrylate function conversion with light excitation for the same amount of time)).

As mentioned above, the benefits of light irradiation on an RIS remains doubtful: in IBA, light irradiation drastically improved polymerization, whereas in EDMA, light irradiation slightly decreased the reactivity. An explanation may be the quantity of inhibitor present in the monomers (900 ppm of 4-methoxyphenol in EDMA, 250 ppm in IBA and 200 ppm in GFMA).

The light absorption properties of thiophenium salts were investigated (available in the Appendix A (Part III)). Weak but significant absorption was observed between 380 and 400 nm, corresponding to one part of the emission spectrum of the LED used during RT-FTIR experiments, in good agreement with the ability of this system to perform photoactivation.

#### 2.2.6. Study of the System Stability

The stability of the system was investigated by aging the two formulations (for both Thiophenium (III) and Na-Tol-sulfinate cartridges) in an oven at 40 °C. The degradation of the system reactivity was followed by optical pyrometry every week during the aging period (Figure 11).

The Thiophenium (III)/Na-Tol-sulfinate system presents excellent stability with almost the same amount of gel time (around 28 s) and the same maximum temperature reached (around 141 °C) before and after aging, with no visible change in the viscosity of the samples.

### 2.3. Mechanistic Consideration

Thiophenium salts are well studied in the field of organic chemistry for their effectiveness in trifluoromethylation reactions [22]. In particular, they are used together with sodium benzenesulfinate to produce phenyltrifluoromethylsulfone with good yields [23]. Although the thiophenium/sodium benzenesulfinate couple has never been used as an RIS, its mechanistic behavior is well known and takes place in three steps [23]: (1) mixing Na-Tol-sulfinate with thiophenium salts results in a counter ion exchange; (2) a single electron transfer occurs, leading to the formation of •CF_3_ and Tol-SO_2_• radicals, and at this stage, •CF_3_ is able to initiate the polymerization steps; whereas (3) Tol-SO_2_• evolves toward the formation of Tol• (Figure 2).

ESR spin trapping experiments were conducted to confirm the proposed mechanism. Simulated spectra revealed the generation of ^•^CF_3_ radical, whereas Tol-SO_2_• or Tol• were not detected. Indeed, the hyperfine coupling constants for the PBN-CF_3_ spin adduct (Figure 12: a_n_ = 14.0 ± 0.1 G; a_H_ = 1.2 ± 0.1 G and a_F_ = 1.7 ± 0.1 G) are in good agreement with the data in the literature [24].

### 2.4. Regulatory Labeling of the New Redox Agents

According to the REACh European regulation, Thiophenium (III) and Na-Tol-sulfinate are both reported as skin and eye irritants (globally harmonized system (GHS) 07). Compared with BPO (GHS 01, 02 and 07) and 4-*N,N*-TMA (GHS 06 and 08), the new RIS, composed of Thiophenium (III)/Na-Tol-sulfinate, can be safer to use, leading to a huge advantage for practical applications.

## 3. Experimental Section

### 3.1. Chemical Compounds

All chemicals were purchased with high purity and used as received (Figure 2, Figure 3 and Figure 4). 2-(2-Aminoethylamino)ethanol (AEAE); *N,N,N’,N’*-tetramethylethylenediamine (TEMED); and *N,N*-dimethyl-p-toluidine (4-*N,N*-TMA) were purchased from TCI-Europe (Bostik, Venette, France). Further, 3,5-Diethyl-1,2-dihydro-1-phenyl-2-propylpyridine (DHPP); 5-(trifluoromethyl)dibenzothiophenium trifluoromethanesulfonate (Thiophenium (III)); and sodium p-toluenesulfinate (Na-Tol-sulfinate) were purchased from Merck. In addition, 2,8-Difluoro-5-(trifluoromethyl)dibenzothiophenium trifluoromethanesulfonate (Thiophenium (II)) was purchased from Combi-Blocks. Dibenzoyl peroxide (BPO) was provided by Bostik as a paste diluted by 50%_wt_ in mineral oil, and 3,7-Ditertiobutyl-5 (trifluoromethyl)dibenzothiophenium trifluoromethanesulfonate (Thiophenium (I)) and 2,8-Difluoro-S-(trifluoromethyl)dibenzothiophenium hexafluorophosphate (Thiophenium IV) were synthetized. The respective efficiencies of the RISs were benchmarked in two acrylate monomers ((5-ethyl-1,3-dioxan-5-yl)methyl acrylate—EDMA; isobornyl acrylate—IBA), provided by Arkema, and one methacrylate resin (glycerol formal methacrylate—GFMA), provided by Bostik.

### 3.2. Two-Cartridge System Configuration for RFRP

Redox formulations were prepared as follows: in a first glass vial, 0.03 g (1%_wt_ based on total monomer mass after mixing) of an oxidizing agent is added to 1.47 g of a (meth)acrylate monomer. In a second glass vial, 0.03 g (1%_wt_ based on total monomer mass after mixing) of a reducing agent is added to 1.47 g of a (meth)acrylate monomer. The two vials are agitated for 2 h at ambient temperature with a magnetic stirrer. Then, the two formulations are placed in a 1:1 Medmix mixpac mixing syringe dispenser (Figure 13).

Polymerization experiments are carried out by mixing the two cartridges at ambient temperature (±25 °C) and dispensing the mixture under air (an oxygen inhibition is then expected, particularly at the surface of the polymer [16]).

### 3.3. Efficiency of the RIS Followed by Optical Pyrometry

Redox polymerization is strongly exothermic, and optical pyrometry is a well-established technique to follow redox polymerization (see [2,13,14,15,16,17]). Therefore, the efficiency of an RIS can be easily followed qualitatively by optical pyrometry. An Omega OS552-V1-6 infrared thermometer (Omega Engineering, Inc., Stamford, CT, USA) with ±1 °C sensitivity was used to determine the temperature vs. time profiles, thus allowing for measuring the gel time associated with the RIS (i.e., the time required after mixing the formulations to transition from liquid to solid state). For simplification, in this work, the gel time will be defined as the time needed by the system to reach the maximum temperature (exothermic peak). The redox system was investigated in a solvent, and no significant increase of temperature was found (<1 °C). This highlights that the temperature increase is related to the monomer conversion, in full agreement with the conversion observed in RT-FTIR spectroscopy (see below).

### 3.4. Real-Time Fourier Transform Infrared (RT-FTIR) Spectroscopy

The conversion of the (meth)acrylate C=C double bond over time was followed by real-time Fourier transform infrared spectrometry (RT-FTIR) using a JASCO 4100 spectrometer (JASCO, Les Lisses, France), as presented by P. Garra et al. [7].

Experiments were conducted by following the (meth)acrylate peak at 6100 cm^−1^ to 6200 cm^−1^ on samples having a thickness of 1 mm. Photopolymerization experiments using an LED emitting at 405 nm with an irradiance of 170 mW.cm^−2^ were also investigated.

### 3.5. Synthesis of Thiophenium Salts

In order to characterize the reactivity of thiophenium salts, some of them were synthetized according to the patent provided by Zhejlang Jiuzhou pharmaceutic [18], as presented in Section 2.1.1 and Section 2.1.2.

All chemicals were purchased with high purity and used as received. First, 4,4′-Di-*tert*-butylbiphenyl, trifluoromethanesulfonic acid, trifluoromethanesulfonic anhydride and nitromethane were purchased from Merck (Merck, Strasbourg, France). Second, sodium hexafluorophosphate, sodium hydrogen carbonate, sodium chloride, acetonitrile, diethyl ether, toluene and dichloromethane were purchased from TCI-Europe.

### 3.6. Electron Spin Resonance (ESR) Spectroscopy

Radical species generated during the reaction of thiophenium salt with sodium p-toluenesulfinate were also studied using ESR spectroscopy (X-band spectrometer, Bruker, EMXplus Biospin, Karlsruhe, Germany).

ESR experiments were performed under oxygen-free atmosphere (nitrogen) in *tert*-butylbenzene (TBB) as a solvent. Concentrations of both thiophenium salt and sodium p-toluenesulfinate were 10^−3^ mol.L^−1^ in TBB. The spin trap used for this study was *N*-*tert*-butyl-α-phenylnitrone (PBN), also used at a concentration of 10^−3^ mol.L^−1^ in TBB. Simulations of the spectra were carried out using the WINSIM software.

## 4. Conclusions

New redox systems have been developed on the basis of the chemistry of thiophenium salts. Good reactivity is observed for thiophenium salts/Na-Tol-sulfinate and thiophenium salts/DHPP systems in mild conditions (under air, at room temperature). By overcoming most of the drawbacks of peroxide-based RISs, the proposed system fulfills important criteria, such as (i) high stability, (ii) high reactivity, (iii) short gel time and (iv) lower toxicity. A synthetic pathway for the thiophenium salts was validated at a small scale. The study of other redox agents will be reported in forthcoming works.

## Data Availability

Data are available from the authors on request.

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
