# Peer review of "Thiophenium Salts as New Oxidant for Redox Polymerization under Mild- and Low-Toxicity Conditions"

_molecules, 2023, doi:10.3390/molecules28020627_

Round 1

Reviewer 1 Report

This manuscript demonstrates the use of Thiophenium salts as a new oxidant for redox polymerization under mild and toxic-free conditions.

1. Grammatical errors need to be corrected throughout the manuscript.
2. The introduction is too short. Only 4-N,N-trimethylaniline is mentioned as an amine used in redox systems. The low stability of the peroxide in the monomers and the toxicity of the aromatic amines are mentioned as the main disadvantages of such a system. Other amines, such as N,N,N',N'-tetramethylethylenediamine, are not mentioned.
3. Description how the conversion was measured will be useful to the reader (not just the reference).
4. Information on monomer conversion and molecular weights of the polymers is lacking. Some additional tests to confirm the activity of the initiator system are needed.
5. The statement in lines 131-133 is not supported by the determination of monomer conversion.

Author Response

The reviewer comments are given in italic. The changes in the MS are highlighted in red.

Responses to reviewers:

Reviewer 1

Comments:

This manuscript demonstrates the use of Thiophenium salts as a new oxidant for redox polymerization under mild and toxic-free conditions.
Answer: We thank the reviewer for the positive comments. All modification in the manuscript have been marked in red color.

Revisions:
1. Grammatical errors need to be corrected throughout the manuscript.
Answer: Special care has been taken for grammatical errors along the manuscript

  1. The introduction is too short. Only 4-N,N-trimethylaniline is mentioned as an amine used in redox systems. The low stability of the peroxide in the monomers and the toxicity of the aromatic amines are mentioned as the main disadvantages of such a system. Other amines, such as N,N,N',N'-tetramethylethylenediamine, are not mentioned.
    Answer: the introduction has been revised. A part of the introduction is dedicated to the choice of the amine class (primary, secondary or tertiary), and the preference for aromatic amines (explaining the choice of 4-N,N-TMA). Also another tertiary aromatic amine is now mentioned, which is toxic free.
  2. Description how the conversion was measured will be useful to the reader (not just the reference).
    Answer: A more detailed description of the RT-FTIR experiments has been added, including thickness of the samples and condition of light irradiation.
  3. Information on monomer conversion and molecular weights of the polymers is lacking. Some additional tests to confirm the activity of the initiator system are needed.

Answer: The newly figure 6 give a better understanding of the (meth)acrylate function conversion over time of the RIS compared to the previous table 1.

In this article, we are mainly focused on the redox initiation in mild conditions. The molecular weights were not evaluated, indeed, we are mainly focused on the Tg (glass transition temperature) of the obtained polymers for adhesive applications. The Tg obtained are very similar to the expected ones.  

  1. The statement in lines 331-333 is not supported by the determination of monomer conversion.

Answer: This part has been updated following the reviewer comments.

Reviewer 2 Report

The authors describe several thiophenium salts for redox polymerization and discuss the influence of multiple factors. However, there are many shortcomings, and we do not recommend publishing.

1. The English should be polished significantly. There are many obvious grammar mistakes and colloquial expressions.

2. More efforts should be made to the production of illustrations.   

3. Rewrite introduction. (1) To supplement the mechanism of Redox Free Radical Polymerization, that is why active free radicals are generated. (2) What is the reason for authors to use Thiophenium Salt as a new oxidant species and what are the advantages and potentials? (3) It is indicated in the manuscript “this work will focus on Thiophenium Salts (Scheme 2) as new oxidant species”, but the Thiophenium Salts is not shown in Scheme 2.

4. In 3.2.1, the authors employ optical pyrometry to evaluate the efficiency of the polymerization. From my viewpoint, the results may be imprecise. Part of the heat is generated from the free radical polymerization process, and other part of the heat comes from redox. More credibly, FTIR or DSC should be carried out to identify the polymerization efficiency. The subsequent analysis should also adjust.

5. In 3.2.2, the analysis of the results appears scattered and illogical, and no regular conclusions are found.

6. In 3.2.5, the effects on light needs further analysis. The three monomers selected by the authors show three results, and the role of photoinitiated polymerization needs to be considered.

7. In 3.3, the study on the mechanism draws on the results of other researches, but lacks the support of their own experimental results.

Author Response

The reviewer comments are given in italic. The changes in the MS are highlighted in red.

Responses to reviewers:

Reviewer 2

Comments:

The authors describe several thiophenium salts for redox polymerization and discuss the influence of multiple factors. However, there are many shortcomings, and we do not recommend publishing.

Answer: We thank the reviewer for the comments. All modification in the manuscript have been marked in red color.

Revisions:

  1. The English should be polished significantly. There are many obvious grammar mistakes and colloquial expressions.

Answer: The English has been polished.

  1. More efforts should be made to the production of illustrations.

Answer: Schemes 1, 2 and 3 have been updated for a better comprehension. Indeed, compounds presented in this study are now divided in 3 schemes: Scheme 3: Structures and names of the (meth)acrylate monomers used in this work ; Scheme 4: Structures and names of reducing agent used in this study ; Scheme 5: Structures and names of the thiophenium salts used in this study.

  1. Rewrite introduction. (1) To supplement the mechanism of Redox Free Radical Polymerization, that is why active free radicals are generated. (2) What is the reason for authors to use Thiophenium Salt as a new oxidant species and what are the advantages and potentials? (3) It is indicated in the manuscript “this work will focus on Thiophenium Salts (Scheme 2) as new oxidant species”, but the Thiophenium Salts is not shown in Scheme 2.

Answer: Introduction has been rewritten: a historical background, the mechanism of RFRP, the choice of the amine, and the interest in thiophenium salts have been added.

  1. In 3.2.1, the authors employ optical pyrometry to evaluate the efficiency of the polymerization. From my viewpoint, the results may be imprecise. Part of the heat is generated from the free radical polymerization process, and other part of the heat comes from redox. More credibly, FTIR or DSC should be carried out to identify the polymerization efficiency. The subsequent analysis should also adjust.

Answer: Indeed, optical pyrometry is a quantitative type of analyze largely used in the literature for redox polymerization (e.g. Crivello’s group). It is a good tool to compare reaction conditions (impacts of concentration, chemical nature, monomer, …) since the setup is easy to handle and readily available compared to DSC or RT-FTIR. An accurate determination of the gel time is possible (not easy to do by RT-FTIR). Moreover, the heat generated from the redox process is insubstantial considering the low amount of redox species used (4.10-3 mol.L-1 for Thiophenium (III) as an example) compared to the heat generated by the polymerization process (a good example is the polymerization of IBA in the manuscript: no increasing temperature because RFRP does not occur and redox process alone does not release enough heat).

Moreover, RT-FTIR experiments (Figure 6) are also given to support the conversion vs. time profiles.

  1. In 3.2.2, the analysis of the results appears scattered and illogical, and no regular conclusions are found.

Answer: The analyses and conclusion associated with the results have been updated for a better clarity.

  1. In 3.2.5, the effects on light need further analysis. The three monomers selected by the authors show three results, and the role of photoinitiated polymerization needs to be considered.

Answer: The new figure 6 gives a better understanding of the role of light on the (meth)acrylate function conversion over time of the RIS compared to the previous table 1. Figure 6 highlights the better final (meth)acrylate function conversion reached and a better rate of polymerization except for EDMA monomer.

To complete the impact of light irradiation, UV-Visible experiments were conducted in order to determine the absorbance of the thiophenium salts (Figure in SI).

  1. In 3.3, the study on the mechanism draws on the results of other researches, but lacks the support of their own experimental results.

Answer: In order to support the mechanism assumptions, Electron Spin Resonance (ESR) experiments were conducted (and added in the manuscript). Results highlight generation of •CF3 radicals.

Reviewer 3 Report

In this study, a new redox initiating system (RIS) based on thiophenium salts as the oxidant species is presented with various reductive species. In addition of the non-toxicity of the developed system, reactivity, stability and synthesis of thiophenium salts were investigated. This manuscript may be accepted after minor modification.

1. The typing mistake on Thiophenium saltsshould be corrected to thiophenium salts in the text, such as 2.5. Synthesis of Thiophenium salts In order to characterize the reactivity of Thiophenium salts, 2,8-Difluoro-5 (Trifluoromethyl)dibenzothiophenium trifluoromethanesulfonate...Trifluoromethyl should be trifluoromethyl...Please carefully check it through the whole text.

2. Please provide references on this point“We assume that this result is mainly due to the structure and/or the additives used with EDMA because this phenomenon does not exist in other monomers as shown in Table 1.”, and why is (meth)acrylate function less efficient with light excitation? Are there any other reasons?

3. What is the maximum absorption wavelength of the new thiophenium salts, and have the absorption spectra of the new thiophenium salts been conducted?

4. Chemical structures of all the involved thiophenium salts should be displayed in the manuscript as a new scheme .

5. It is better to add a discussion on the main difference on Thiophenium (I)~(IV) in Conclusionsection to understand the structure-property relationships in practical application.

Author Response

The reviewer comments are given in italic. The changes in the MS are highlighted in red.

Responses to reviewers:

Reviewer 3:

Comments:

In this study, a new redox initiating system (RIS) based on thiophenium salts as the oxidant species is presented with various reductive species. In addition of the non-toxicity of the developed system, reactivity, stability and synthesis of thiophenium salts were investigated. This manuscript may be accepted after minor modification.

Answer: We thank the reviewer for the positive comments. All modification in the manuscript have been marked in red color.

Revisions:

  1. The typing mistake on “Thiophenium salts”should be corrected to “thiophenium salts” in the text, such as “2.5. Synthesis of Thiophenium salts In order to characterize the reactivity of Thiophenium salts,” “2,8-Difluoro-5 (Trifluoromethyl)dibenzothiophenium trifluoromethanesulfonate”...Trifluoromethyl should be trifluoromethyl...Please carefully check it through the whole text.

Answer: Names have been updated in accordance with the literature.

  1. Please provide references on this point “We assume that this result is mainly due to the structure and/or the additives used with EDMA because this phenomenon does not exist in other monomers as shown in Table 1.”, and why is (meth)acrylate function less efficient with light excitation? Are there any other reasons?

Answer: Industrial FDS of monomers reveal that in the case of EDMA, the content of inhibitor (MEHQ) is the highest. Another explanation may be the color of the monomer: indeed, EDMA has a yellow color (so absorbing in the blue/purple) whereas IBA and GFMA have no color (transparent).

  1. What is the maximum absorption wavelength of the new thiophenium salts, and have the absorption spectra of the new thiophenium salts been conducted?

Answer: To have a better understanding of the light absorption properties of thiophenium salts, new experimental data are available in the “3.2.5. Effect of light activation” section.

  1. Chemical structures of all the involved thiophenium salts should be displayed in the manuscript as a new scheme.

Answer: Scheme 1, 2 and 3 of the previous manuscript have been updated for a better comprehension. Indeed, compounds presented in this study are now divided in 3 schemes: Scheme 3: Structures and names of the (meth)acrylate monomers used in this study ; Scheme 4: Structures and names of reducing agent used in this study ; Scheme 5: Structures and names of the thiophenium salts used in this study.

  1. It is better to add a discussion on the main difference on Thiophenium (I)~(IV) in “Conclusion”section to understand the structure-property relationships in practical application.

Answer: A small conclusion at this end of the section “3.2.2 effect of the oxidizing agent” has been added to discuss about the best possible structure/reactivity relationship for thiophenium salts

Round 2

Reviewer 2 Report

The authors have made a comprehensive revision of the article, but there is still a question that the author needs to answer. Please design experiments to verify the heat generated from the redox process is insubstantial in the optical pyrometry.

Author Response

The comments are given in italic. The changes in the MS are highlighted in red.

Responses to reviewers:

Reviewer #2:

Recommendation: Publish after minor revisions.

Comments: “The authors have made a comprehensive revision of the article, but there is still a question that the author needs to answer. Please design experiments to verify the heat generated from the redox process is insubstantial in the optical pyrometry..

Answer: We thank the reviewer for the very positive comments.

Issue Raised:

“there is still a question that the author needs to answer. Please design experiments to verify the heat generated from the redox process is insubstantial in the optical pyrometry.”

Answer: We thank the reviewer for the question. Optical pyrometry is a well-established technique to follow redox polymerization (see ref [2,13-15,17]). These references are now given in the experimental part. Indeed, the polymerization process is extremely exothermic (78.6 kJ/mol released for acrylates and 52 kJ/mol for methacrylate functions). Here, the resin is composed of 98% of polymerizable functionalities. To highlight this behavior, the redox system was investigated in a solvent (acetonitrile) and no significant increase of temperature is found (< 1°C). This highlights that the temperature increase is related to the monomer in full agreement with the conversion observed in RT-FTIR spectroscopy!
